# Production of 5-Hydroxymethylfurfural from Chitin Biomass: A Review

**DOI:** 10.3390/molecules25030541

**Published:** 2020-01-27

**Authors:** Dan Zhou, Dongsheng Shen, Wenjing Lu, Tao Song, Meizhen Wang, Huajun Feng, Jiali Shentu, Yuyang Long

**Affiliations:** 1Zhejiang Provincial Key Laboratory of Solid Waste Treatment and Recycling, Analysis and Testing Center, School of Environmental Science and Engineering, Zhejiang Gongshang University, Hangzhou 310012, China; dddddd926@163.com (D.Z.); shends@zju.edu.cn (D.S.); wmzyy@163.com (M.W.); fenghuajun2000@163.com (H.F.); shentujl@mail.zjgsu.edu.cn (J.S.); 2School of Environment, Tsinghua University, Beijing 100084, China; luwenjing@tsinghua.edu.cn; 3School of Energy and Mechanical Engineering, Nanjing Normal University, Nanjing 210023, China; tsong@njnu.edu.cn

**Keywords:** 5-hydroxymethylfurfural, chitin biomass, pretreatment, chitosan, degradation, biorefinery

## Abstract

Chitin biomass, a rich renewable resource, is the second most abundant natural polysaccharide after cellulose. Conversion of chitin biomass to high value-added chemicals can play a significant role in alleviating the global energy crisis and environmental pollution. In this review, the recent achievements in converting chitin biomass to high-value chemicals, such as 5-hydroxymethylfurfural (HMF), under different conditions using chitin, chitosan, glucosamine, and *N*-acetylglucosamine as raw materials are summarized. Related research on pretreatment technology of chitin biomass is also discussed. New approaches for transformation of chitin biomass to HMF are also proposed. This review promotes the development of industrial technologies for degradation of chitin biomass and preparation of HMF. It also provides insight into a sustainable future in terms of renewable resources.

## 1. Introduction

Investigation of green renewable energy has gradually become a main research direction in recent years. As a substitute for fossil fuel, biomass is a potential renewable resource for future production of fuels and high value chemical products owing to the vast number of organic chemical products that can be potentially manufactured from biomass [1,2]. Biomass includes plant fibers, such as lignocellulose, and animal fibers, such as chitin [3]. Lignocellulosic materials have been widely investigated [4,5]. Chitin is the world’s second most abundant biopolymer. However, chitin biomass transformation is still in the initial stage. Chitin biomass includes raw materials, such as crustacean shells, and purified chitin, chitosan, and its corresponding monomers (*N*-acetylglucosamine (GlcNAc) and glucosamine (GlcNH_2_)) and derivatives. Chitin widely exists in crustacean shells, insect skeletons, fungi, etc. With the rapid growth of the fishery industry, which is the main source of chitin biomass, global fishery production has become well above 100 million tons per year with a large amount of crustacean wastes. An estimated 6–8 million tons of shell waste from crustaceans is produced annually [6]. Generally, most crustacean waste is directly disposed of in landfills without utilization which not only poses environmental issues, but it is also a waste of resources [7]. Valorization of shell waste will have both ecological and economic benefits.

Determining the potential value of chitin biomass is in accordance with the concept of green chemistry. There is a growing interest in chitin biomass, and conversion of chitin biomass to high value-added compounds has become a new research hotspot. In recent years, the concept of shell biorefinery has been proposed [3,8,9]. Chitin biomass should be fully utilized like cellulose biomass by converting it to high value chemicals (Figure 1). Among the various high-value chemicals, 5-hydroxymethylfurfural (HMF) is one of the most promising value-added platform compounds [10]. 5-Hydroxymethylfurfural, which is highly chemically reactive, can be prepared by oxidation, hydrogenation, etc. It has huge market potential which can be used to produce 2,5-furandialdehyde, 2,5-furandicarboxylic acid, levulinic acid (LA), and other high value-added chemicals and liquid fuels. Chitin biomass is an ideal raw material for preparation of HMF. Tapping the potential of chitin biomass to obtain the high value-added platform compound HMF is an important way to improve the output of waste resources, and it is expected to lead to new directions for high-value utilization of chitin biomass.

This paper reviews the recent related research progress in the conversion of chitin biomass to HMF and summarizes the methods and pathways during the conversion process under different conditions. Finally, the potential future research directions are also proposed. 

## 2. Structure of Chitin Biomass

Chitin biomass includes raw materials, such as shrimp and crab shells, and purified chitin, chitosan, and its corresponding monomers (GlcNAc and GlcNH_2_) and derivatives [11]. Chitin widely exists in nature including in the shells of insects and crustaceans (e.g., shrimps and crabs). The skeletal shells of mollusks and cell walls of certain algae and fungi also contain chitin. It is the most abundant natural nitrogen-containing polymer polysaccharide in the world, and it is also an important nitrogen source for typical earth and marine organisms. 

Chitin is a linear polymer polysaccharide formed by condensing *N*-acetylglucosamine with β-1,4-glucoside bonds, and the relative molecular weight of chitin ranges from hundreds of thousands to millions DA. The chemical structure of chitin is basically the same as lignocellulose. The difference between them is that the functional group at the C_2_ position of cellulose is the hydroxyl group (–OH) while that of chitin is the acetylamino group (CH_3_CONH–), as is shown in Figure 2. Therefore, chitin is sometimes referred to as animal cellulose. The monomer of chitin is GlcNAc which can be obtained from chitin by hydrolysis with concentrated hydrochloric acid (HCl) or sulfuric acid (H_2_SO_4_) or the enzymatic hydrolysis. 

The product obtained after removing most of the *N*-acetyl groups from the sugar group in the chitin structure is called chitosan which is also known as deacetylated chitin. The deacetylated products that have a degree of deacetylation (DD) greater than 60% or are soluble in dilute acid are generally called chitosan. Compared with chitin, chitosan contains less acetyl groups, more amino groups, and has higher solubility. Chitosan has great application potential because of its unique properties and physiological activities. The monomer of chitosan is GlcNH_2_ which is the compound obtained by replacing –OH at the C_2_ position of glucose with –NH_2_, as is shown in Figure 2. It can also exist in various forms in living organism polysaccharides and conjugated polysaccharides.

## 3. Chitin Biomass Pretreatment Techniques

As the main source of chitin, crustacean shells normally contain 20–50% calcium carbonate, 20–40% protein, and 15–40% chitin. The chitin in crustacean shells is embedded and solidified by a mineral–protein matrix [12]. Therefore, it is not easy to directly transform chitin biomass to HMF. Some researchers have found that pretreatment of chitin biomass before the reaction, such as transforming chitin or chitosan to high-purity chitin or low-molecular-weight chitosan, can effectively improve the HMF yield. The conversion process of chitin biomass to HMF is shown in Figure 3. The chitosan reaction activity increases with decreasing molecular weight, and conversion to HMF is also enhanced. Through various pretreatment methods, demineralization and deproteinization are required to remove CaCO_3_ and protein in chitin biomass in order to prepare purity chitin. Furthermore, the acetyl group is removed to convert chitin into low-molecular-weight chitosan and even its monomer by deacetylation. These are conducive to the progress of the reaction. Thus, pretreatment provides favorable conditions for conversion of chitin biomass to HMF. Pretreatment methods for chitin biomass are mainly divided into chemical methods, biological methods, and physical-assisted methods. The principles for choosing a pretreatment method are to minimize the energy consumption and provide good compatibility with the following process. 

### 3.1. Chemical and Biological Pretreatment

Traditional chemical methods usually use dilute HCl solution (0.3–2 M) for demineralization. It consists of eliminating CaCO_3_ and CaCl_2_ which constitute the main inorganic compounds of crustacean shells. During the digestion reaction, the emission of CO_2_ gas indicates the content of mineral materials. Furthermore, deproteinization is performed using alkaline treatment using dilute sodium hydroxide (NaOH) solution (0.5–4 M) to remove proteins. The product obtained is designated as purified chitin. Extraction of chitosan usually requires a deacetylation process in a high-concentration alkali solution at high temperature. Despite the many disadvantages of chemical methods, the extraction time is short which makes it the most commonly used commercial treatment. However, these methods inevitably generate a large amount of corrosive acid–base wastewater which may cause serious environmental problems [13,14].

Biological treatments, such as enzyme reaction and microbial fermentation, offer an alternative way to extract chitin and chitosan [15,16]. For instance, lactic-acid-producing bacteria and proteases from bacteria have been used for the demineralization and deproteinization steps, respectively. Chitin deacetylation have been performed by enzymatic methods. Using these methods, the above problem of environmental pollution can be avoided. Moreover, the molecular weight and crystallinity of the obtained chitin are higher than those of chemically prepared chitin [17]. However, these biological processes are insufficient to remove the minerals and proteins from crustacean shells [18]. In addition, long fermentation cycles and expensive enzymes prevent commercial application of these methods.

As a more sustainable alternative to volatile organic solvents, ionic liquids (ILs) such as 1-allyl-3-methylimidazolium bromide and 1-ethyl-3-methylimidazolium acetate have been used for direct extraction of chitin from crustacean shells [18,19]. Ionic liquids are considered to be promising solvents for chitin production because of their low vapor pressures, non-flammability, and excellent solubilities [20]. They can also promote the deacetylation process [21]. Nevertheless, ILs have some drawbacks, such as high cost, toxicity, and non-biodegradability, which significantly limit their use in various applications [22,23]. Therefore, a more economic, efficient, and eco-friendly technique is desirable for extraction of chitin and chitosan.

Deep eutectic solvents (DESs) are a greener alternative to conventional ILs. They are obtained by mixing a hydrogen bond acceptor and a hydrogen bond donor that are capable of self-association through hydrogen-bonding interactions to form a eutectic with a lower melting point than each individual component [24,25]. Compared with traditional ILs, DESs possess more advantages such as low toxicity, low cost, ease of syntheses, biodegradability, and negligible volatility. Owing to these properties, DESs have various applications including organic synthesis, dissolution media, extraction processes, and materials chemistry [26]. Some DESs prepared from choline chloride and four organic acids have been evaluated for the extraction of chitin from lobster shell; among them, the purity of chitin extracted with choline chloride-malonic acid was the highest [27,28]. Furthermore, natural DESs (NADESs), which are composed of natural compounds, are sustainable, non-toxic, and biodegradable. Choline chloride–malic acid has been used to extract chitin, and most of the minerals and proteins in shrimp shells can be removed under microwave-assisted conditions. The NADESs possess better properties during extraction than ILs and DESs [29]. However, the ideal type of DES for biomass conversion is still unclear. In addition, there is a lack of product diversity in the downstream conversion process of DES-fractionated products and further research is needed [30].

Some researchers are already investigating the use of natural resources, such as glycerol, as the reaction solvent for extraction of chitin and chitosan as is shown in Table 1. Glycerol, a by-product of biodiesel production, is readily available, non-toxic, inexpensive, biodegradable, and has deproteinization ability. Recent studies have shown that acids are capable of deproteinizing chitin during extraction of chitin from crustacean shells [31]. Therefore, some researchers have combined an acid and glycerol to produce chitin. Hong et al. [32] used a glycerol and HCl co-solvent to achieve demineralization and deproteinization at low temperature for one-step production of chitin from lobster shells. Although HCl was used in this method, the acid amount was lower than that required for the conventional method. Devi et al. [33] used citric acid instead of HCl to directly demineralize and deproteinize chitin after preliminary deproteinization by hot glycerol pretreatment. Plasticization of prawn shell waste by glycerol enabled uniform heat transfer, and it could possibly be the driving force for thermal depolymerization of proteins in the absence of acid or base catalysts. In addition, the glycerol used in the pretreatment was effectively reused and recycled. Using citric acid is also more environmentally friendly than using HCl. Using glycerol as the reaction solvent for deacetylation of chitin could realize deacetylation of chitin with low NaOH concentration and provide an efficient and green process for chitosan extraction from chitin [34]. 

Recently, Yang et al. [35] developed a new economical and feasible shrimp shell fractionation method using hot water deproteinization and carbonic acid demineralization to extract high-value chitin from crustacean shells such as shrimp shells. The proteins were partially hydrolyzed and fully solubilized in water at high temperature. The CaCO_3_ was then dissolved under pressured CO_2_ in aqueous solution at room temperature. This simple and feasible method can give chitin products with high purity (>90%) while reducing costs and waste generation.

### 3.2. Physical-Assisted Pretreatment

Because some of the above methods are time consuming and energy intensive, a wide variety of physical-assisted techniques for production of low-molecular-weight chitin or chitosan have been investigated such as the ultrasonic, microwave-assisted alkali, mechanochemical, and plasma treatment [36].

Some researchers have investigated ultrasonic pretreatment of chitosan which can increase the HMF yield. Other researchers have performed ultrasound-assisted deacetylation to convert both α-chitin and β-chitin to chitosan [37,38]. High-intensity ultrasound irradiation strongly enhances the *N*-deacetylation reaction, favoring production of completely acid-soluble chitosan at high yield. In addition, the ultrasound-assisted deacetylation process allows preparation of unusually high molecular weight randomly deacetylated chitosan. Ultrasound-assisted extraction of chitosan can also significantly reduce the molecular weight and improve the properties of chitosan [39].

In the past decade, microwave irradiation has been widely used as a powerful tool for rapid and efficient synthesis. This new technology has replaced conventional heating using three-dimensional heating of the reaction mass [40,41]. Many researchers have successfully deacetylated chitin to chitosan using microwave radiation, with the DD reaching 80–95% in a few minutes and the molecular weight greatly decreasing. However, microwave deacetylation still consumes as much NaOH as traditional deacetylation [42,43,44,45]. El Knidri et al. [46] were the first to use microwave technology for demineralization and deproteinization. Microwave radiation was used throughout. It shortened the duration time to 1/16 of that of the traditional method. The chitosan produced by microwave heating had the same DD as the traditional method (DD of chitosan produced by conventional heating 81.5% and DD of chitosan produced by microwave heating 82.7%). Compared with conventional extraction, microwave synthesis dramatically decreases the reaction time and increases the product yield and purity by decreasing unwanted side reactions. 

Mechanochemistry that uses mechanical energy to activate chemical reactions is also widely used in high value-added conversion of various barely degradable biomasses [47,48,49]. Chen et al. [50] achieved one-step conversion of chitin and raw shrimp shells to low-molecular-weight chitosan in the solid state by using the synergistic mechanical force and base catalysis. Ball milling significantly and simultaneously enhanced base-catalyzed depolymerization and deacetylation of chitin, giving a chitosan oligomer with 80.5% DD and 7.9kDa molecular weight. The base usage was reduced to about 1/10 of that used by traditional approaches. The synergistic effects of the mechanical and chemicals forces were crucial for simultaneous depolymerization and deacetylation. However, treatment takes about 2 h which is much longer than the microwave deacetylation time, so energy consumption is relatively high. 

Green chemistry is an important issue in current society, and a green and feasible pretreatment method should be used for biomass conversion. An efficient green pretreatment extraction process of chitin and/or chitosan should be selected by combining reasonable pretreatment methods to realize the possibility of chitin biomass conversion to HMF. For instance, to promote the reaction, suitable green solvents, such as DESs and glycerol, can be added to the catalytic reaction system. Furthermore, physical-assisted methods can be combined with chemical or biological methods to take advantage of their respective advantages. For example, green solvents, such as DESs [51] or glycerol, could be used in microwave processes. Microwave radiation not only shortens the reaction time for extracting chitin and chitosan, but it also greatly reduces the amount of acid or alkali in the reaction with the addition of green solvents. Through efficient and green extraction of chitin and chitosan, conversion of chitin biomass to HMF can be achieved.

## 4. HMF Production from Chitin Biomass

5-Hydroxymethylfurfural has been ranked as the most valuable bio-based platform compound derived from biomass [10]. It is one of the most important high value-added compounds [52], and it is widely used in the preparation of multifunctional compounds such as fine chemicals, key pharmaceutical intermediates, functional polyesters, solvents, and liquid fuels [53]. It can be obtained by dehydration of glucose and fructose produced by hydrolysis of lignocellulose [54], as shown in Figure 4. Chitin is similar to cellulose in structure, so chitin biomass can also be converted to HMF. Although conversion of chitin to nitrogen-free high value-added chemical-waste nitrogen resources, production of nitrogen-free platform compounds provides more choices for the effective use of biomass, so it is easier to achieve biorefinery. Therefore, conversion of chitin biomass to HMF has important research significance.

### 4.1. HMF Production from Chitin Biomass by Brønsted Acids

Brønsted acid is a general term for molecules or ions that release protons. Common Brønsted acids (e.g., HCl, H_2_SO_4_, and acetic acid) have low production costs, are easy to prepare, and are widely used. In early studies, HCl solution was usually used to catalyze degradation of chitin biomass, but only trace amounts of HMF were obtained. Omari et al. [55] used HCl to catalyze degradation of chitosan at 200 °C for 30 min under microwave heating, but they only obtained 2.2% HMF. Lee et al. [56] subsequently used 2.2% H_2_SO_4_ instead of HCl to catalyze degradation of chitosan at 174 °C for 36.9 min and obtained 12.1% HMF. Compared with earlier studies, this method not only gave higher yield, but it also significantly reduced the reaction temperature and acid concentration. Therefore, energy consumption and waste acid production were significantly reduced. Jeong et al. [57] also catalyzed GlcNH_2_ degradation under similar conditions to those mentioned above. They achieved an HMF yield of 1.8% in 5 min, but the HMF yield decreased with time. Under similar conditions, the type and concentration of the substrate greatly affected the HMF yield. It is well known that glucose isomerizes to fructose and then dehydrates to form HMF. As with glucose, GlcNH_2_ in the form of pyranose first isomerizes to furanose. Subsequently, GlcNH_2_ removes the NH_2_ groups in furanose by deamination under acidic conditions. Finally, HMF is formed by dehydration and keto-enol tautomerization, and it then rehydrates to levulinic acid (LA) and formic acid. Therefore, GlcNH_2_ as the monomer of chitosan does not easily form HMF under the above conditions, but it has the potential to convert to LA, so the HMF yield is low. 

Methanesulfonic acid is strong, non-oxidizing, and biodegradable, indicating that it causes fewer environmental problems than inorganic acids. Moreover, it is an attractive alternative to organic and inorganic strong acids in various applications such as biomass valorization [58,59]. Jeong et al. [60] used 0.1 M methanesulfonic acid to catalyze 100 g/L GlcNH_2_ at 160 °C for 40 min, but only 2.3% HMF was produced. With increasing temperature, the HMF yield continued to decrease, because GlcNH_2_ was converted to HMF and further hydrolyzed to LA. They also used 2% chitosan as a substrate under the same catalytic conditions at 200 °C for 15 min, and the HMF yield reached 15.0% [61] which far exceeded the yield in their previous study. This further validated the above hypothesis that the small molecular weight and simple structure of GlcNH_2_ make it easier to convert it to LA than chitosan. Therefore, the type and concentration of the substrates should be carefully selected to improve the HMF yield. According to the substrates, the chitin biomass pretreatment method should also be carefully chosen.

Savitri et al. [62] introduced ultrasound-assisted technology to the acetic-acid-catalyzed reaction system, and all of the partially dissolved chitosan was dissolved in the acetic acid solution. After chitosan was sonicated at 40 °C for 30 min in low concentration (0.5% *v*/*v*) acetic acid solution, a higher concentration of HMF was detected in the soluble chitosan component. However, under the same conditions, chitosan degraded to GlcNH_2_ as the main product in high concentration acetic acid solution (1% *v*/*v*). Therefore, the catalyst concentration is also a critical factor in conversion of chitin biomass to HMF. 

Although Brønsted acids have the advantages of large output and simple operation, the use of a high concentration of a corrosive acid (e.g., HCl and H_2_SO_4_) will seriously corrode the reaction equipment, pollute the environment, and increase the possibility of by-product formation. This makes separation of the target product difficult and high-value conversion of chitin biomass tedious and expensive. Although organic Brønsted acids are less corrosive, they are slightly toxic which also limits development and research of organic Brønsted acid catalysts to a certain extent. Therefore, selecting an efficient catalytic system and optimizing the reaction conditions to improve the HMF yield still require more in-depth research. 

### 4.2. HMF Production from Chitin Biomass by Lewis Acids

Conversion of chitin biomass to HMF by Brønsted acids has some disadvantages such as complex post-treatment, low target product yield, and intractable pollution. Therefore, an efficient catalytic system for degradation of chitin biomass is urgently required. Lewis acids are substances, including ions, radicals, and molecules, that can accept electron pairs. Compared with Brønsted acids, Lewis acids have more obvious advantages, such as low cost and high catalytic activity. Therefore, the process of Lewis acid (represented by metal chlorides) catalyzed degradation of chitin biomass to HMF has been intensively investigated.

Omari et al. [55] found that Lewis acids have higher catalytic efficiency than Brønsted acids. Among the Lewis acids investigated, SnCl_4_·5H_2_O showed the best catalytic effect. When chitosan was heated at 200 °C under microwave radiation for 30 min with a low concentration of SnCl_4_·5H_2_O, 10.0% HMF was obtained. When chitosan was similarly treated with a higher concentration of SnCl_4_·5H_2_O, no HMF was produced, but up to 12.7% LA was obtained. It is possible that high catalyst loading not only accelerates conversion of chitosan to HMF, but it also promotes other side reactions, such as rehydration of HMF to LA and cross-polymerization reactions, which lower the HMF yield. Therefore, a suitable catalyst concentration should be chosen to effectively increase the target product yield. Compared with traditional heating, microwave heating has the advantages of fast heating, uniform heating, and short reaction time. Therefore, microwave heating has been applied to obtain HMF (traditional heating does not generate HMF). However, the HMF yield was still low and needs to be further optimized by changing the metal salt type. 

Deng et al. [63] found that concentrated ZnCl_2_ solution not only dissolves lignocellulose, but it also shows a good catalytic effect for degradation in lignocellulosic biomass hydrolysis. Incompletely coordinated Zn^2+^ can coordinate with the –OH groups of carbohydrates (e.g., cellulose), thereby catalytically converting them to fructose and then to HMF. Based on the structural similarity of cellulose and chitin, there is also a strong interaction between Zn^2+^ and the –OH, –NH_2_, and/or –NHAc groups in chitin biomass [64,65]. Owing to protonation of these groups, the solubility of chitin biomass in concentrated ZnCl_2_ solution greatly increases. Wang et al. [66] used 67% concentrated ZnCl_2_ solution to catalyze GlcNH_2_ conversion at 120 °C for 90 min, and they obtained 21.9 mol% HMF. Therefore, concentrated ZnCl_2_ solution is an environmentally friendly and inexpensive chitin biotransformation medium. 

There are clear differences in the catalytic effects of metal salts with different metal ions. Yu et al. [67] investigated the effects of 16 different metal salt catalysts on conversion of chitosan and GlcNAc to HMF, including AlCl_3_, MgCl_2_, FeCl_2_, FeSO_4_, and the abovementioned ZnCl_2_ and SnCl_4_. Among these catalysts, FeSO_4_·7H_2_O (22.8 mol%) and FeCl_2_·4H_2_O (33.9 mol%) showed high catalytic activity in conversion of GlcNAc to HMF. Reacting chitosan in dimethyl sulfoxide (DMSO)–water-mixed solvent at 190 °C in FeCl_2_·4H_2_O for 6 h gave an HMF yield of 26.6 mol%. This high yield is caused by the excellent solubility and long reaction time of the DMSO–water-mixed solvent. The use of a suitable solvent with the catalyst enables the catalyst to better dissolve in the solvent, thereby improving the HMF yield. 

Sulfamic acid (NH_2_SO_3_H) has a tautomeric structure with Brønsted acid and Lewis acid sites. As a dual catalyst, it has unique catalytic properties and inherent zwitterionic properties. Therefore, it is often used as a substitute for conventional acid catalysts to discover new chemical reactions and processes [68]. Some studies have used NH_2_SO_3_H to catalyze degradation of lignocellulose [69], and Jeong et al. [70] successfully applied NH_2_SO_3_H to chitin biomass conversion to HMF for the first time. They used 0.7 M NH_2_SO_3_H to catalyze 3% (*w*/*w*) chitosan conversion at 200 °C, achieving a high yield of 21.5% in 2 min. Compared with previous studies, higher HMF yields were achieved in a much shorter reaction time, showing the extraordinary catalytic effect of NH_2_SO_3_H.

The common disadvantages of homogeneous acid–base catalysts are many by-products, severe equipment corrosion, and complex post-treatment processes. Heterogeneous acid catalysts, also known as solid acid catalysts, are compounds with catalytically active acidic sites on the solid surface. They are easy to separate and recover from the product and can be easily recycled. They are also environmentally friendly and only slightly damage the reactor. Many homogeneous acid–base catalysts have been gradually replaced by solid acid–base catalysts. Solid acid–base catalysts are an important direction for future catalytic development. When designing a solid acid catalyst, the number of strong acid sites needs to be optimized to improve the catalytic activity which is beneficial for production of HMF. In addition, the type of acid sites in the solid acid catalyst is also an important factor that determines the reaction pathway. Common solid acid catalysts include carbon-based solid acids, molecular sieves, ion-exchange resins, and heteropoly acids [71]. Degradation of monosaccharide, starch, cellulose, and other biomasses to HMF using solid acid catalysts has been investigated (Table 2) [72,73,74,75,76], but degradation of chitin biomass using solid acid catalysts has rarely been reported. Kalane et al. [77] combined a solid acid catalyst with different concentrations of acetic acid to catalyze chitosan conversion to HMF by “green” synthesis. The H-zeolite catalyst combined with the lowest concentration of acetic acid produced the highest yield of HMF (15.3–28.2%). There are many types of solid acids. Therefore, screening solid acid catalysts for conversion of chitin biomass to HMF is an important direction for future research. 

### 4.3. HMF Production from Chitin Biomass Using ILs

In addition to the catalyst, the solvent is also crucial for degradation of chitin biomass. The solvent can not only dissolve the substrate and catalyst to promote the reaction, but it can also stabilize the thermodynamic equilibrium of the substrate, intermediates, and products to obtain a high product yield and be used as a catalyst to improve the reaction kinetics. Iion liquids have attracted widespread attention as new environmentally friendly “green solvents”. They have many advantages such as good stability, low melting point, low vapor pressure, adjustable acidity, good density, viscosity, solubility, and recyclability. They can be used as a solvent and a catalyst in the biomass conversion process to participate in chemical reactions, and they have the advantages of high conversion rate and good selectivity. They are widely used in chemical extraction and separation, organic synthesis, electrochemistry, nanomaterials, and other new functional materials, and they are becoming one of the most promising reaction media and catalytic systems in green chemistry [78,79,80].

A lot of studies have reported the excellent performance of ILs in catalyzing degradation of cellulose biomass or monosaccharides to produce HMF [80,81,82,83]. Use of IL catalytic systems to degrade chitin biomass has also attracted interest. By introducing different functional groups to the anions and cations, different functionalized ILs have been designed to optimize the catalytic effect. Investigating the reaction conditions is also crucial to optimize the HMF yield from chitin biomass. 

In recent years, considerable effort has been devoted to catalytic degradation of chitin biomass in ILs. Feng et al. [84] used 3-methyl-1-butylimidazole hydrogen sulfate IL as the catalyst and AlCl_3_⋅6H_2_O as a cocatalyst for catalytic degradation of chitosan at 180 °C for 5 h, and they achieved a HMF yield of 25.2 mol%, providing the concept of synergistic catalytic degradation of chitin biomass by Lewis acids and ILs. Li et al. [85] chose nine types of acidic ILs with different anions and cations for catalytic degradation of chitosan, and *N*-methylimidazole hydrogen sulfate ([MIM]HSO_4_) IL showed the best catalytic effect. They also found that the acidity, hydrogen bonding ability, and steric hindrance of the IL affected the reaction. Using [MIM]HSO_4_ as a catalyst to catalyze conversion of chitosan and chitin at 180 °C for 5 h, they achieved HMF yields of 29.5 and 19.3 mol%, respectively. However, they believed that the highest HMF yield for chitin conversion of 19.3 mol% was still low. Further research is therefore required to develop an efficient and economically feasible HMF production process from chitin biomass. Zang et al. (Table 3) [86] found that methyl imidazole hydrogen sulfate ([Hmim][HSO_4_]) IL showed the best catalytic performance for chitosan conversion at 180 °C for 6 h in DMSO–water-mixed solvent among seven acidic ILs, with a HMF yield of 34.7 mol%. The catalytic effect of ILs has been improved by improving the type of IL. To achieve conversion of chitin biomass to HMF under mild conditions, Zhang et al. [87] used benzimidazole chloride ([Hbim]Cl) IL to degrade chitosan at 180 °C for 3 h in 10 wt% DMSO–water-mixed solvent. They achieved a highest HMF yield of 34.9%, while the HMF yield in the pure water system was 30.8 mol%. It is clear that the type of IL and the choice of DMSO solvent are significant. Owing to the improved catalytic effect of ILs, the reaction time to achieve the same HMF yield was reduced by half (3 h) compared with the reaction in the pure water system (6 h). This indicates that [Hbim]Cl is an efficient, sustainable, and environmentally friendly catalyst for conversion of chitin biomass to high value-added chemicals. Additionally, as environmentally friendly catalysts, Brønsted–Lewis acidic ILs have also attracted attention, because they combine the characteristics of Brønsted–Lewis solid acids and ILs. Jiang et al. [88] were the first to use a Brønsted–Lewis acidic IL ([Hmim][HSO_4_]–0.5FeCl_2_) to degrade chitosan to HMF. After reaction of chitosan at 180 °C for 4 h under hydrothermal conditions, the HMF yield was 44.1 mol%. They believed that these research achievements will benefit the economy and environment. Therefore, development of new low cost and high activity functionalized ILs is one of the main directions in this field. In addition, new functionalized ILs can also work synergistically with catalysts, such as Lewis acids, to combine their respective advantages.

## 5. Conclusions

In recent years, to reduce the dependence on traditional fossil resources in the chemical industry, attention has turned to renewable biomass resources with high annual output. Preparation of fuels and precursor compounds from biomass has attracted widespread attention. Chitin biomass is the second largest natural renewable biomass resource after cellulose. Conversion of chitin biomass to HMF has great application prospects. However, degradation of chitin biomass is a very complicated multi-step reaction process with many side reactions, heterogeneous products, and difficult separation. Conversion of chitin biomass to HMF still faces many problems. In future research, researchers should focus on the following aspects:

(1) Selection of green and feasible pretreatment methods during biomass conversion. Through selection and combination of suitable pretreatment methods, an efficient and green pretreatment extraction process for chitin and/or chitosan should be developed to realize the possibility of chitin biomass conversion to HMF;

(2) New, highly selective, highly active, and green catalytic solvent systems should be designed to develop green and efficient catalytic systems. The synergistic catalytic effect of different catalysts should also be explored to improve the targeted conversion efficiency of HMF;

(3) The mechanism of HMF conversion should be deeply investigated, and the reaction conditions should be optimized to improve the HMF yield.

## Figures and Tables

**Figure 1 molecules-25-00541-f001:**
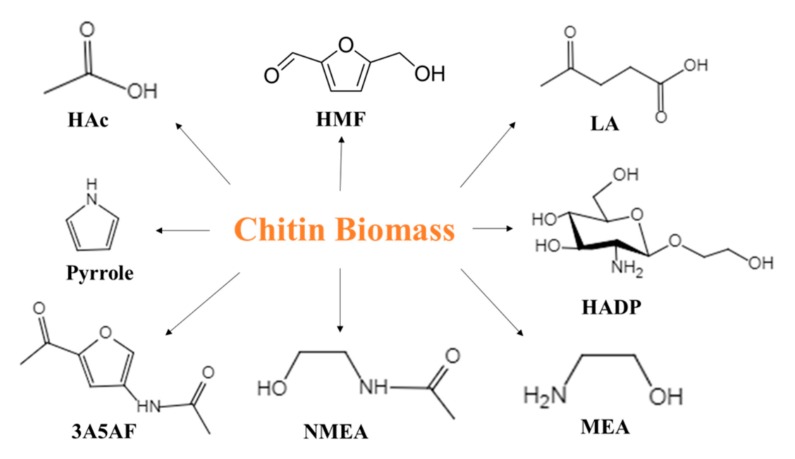
An overview of chitin biomass conversion into various chemicals. HAc, acetic acid; HMF, hydroxymethyl furfural; LA, levulinic acid; HADP, hydroxyethyl-2-amino-2-deoxyhexopyranoside; 3A5AF, 3-acetamido-5-acetylfuran; NMEA, N-acetylmonoethanolamine; MEA, monoethanolamine.

**Figure 2 molecules-25-00541-f002:**
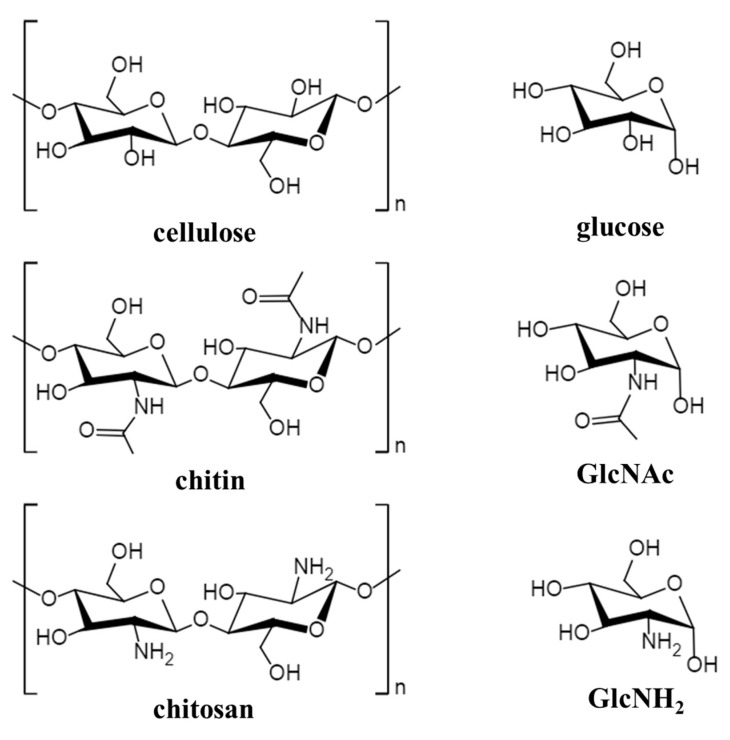
Chemical structure of cellulose, chitin, and chitosan and their corresponding monomers.

**Figure 3 molecules-25-00541-f003:**
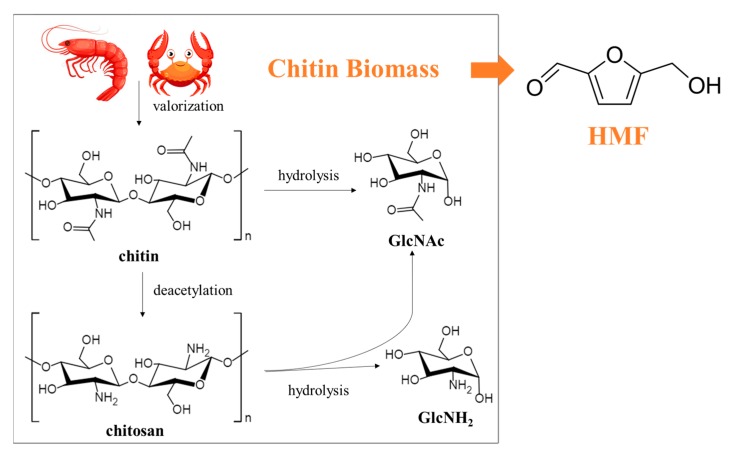
Production of HMF from chitin biomass.

**Figure 4 molecules-25-00541-f004:**
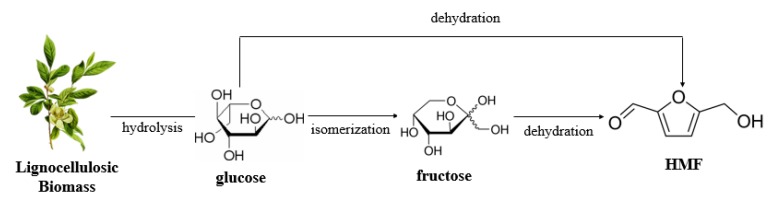
Production of HMF from lignocellulosic biomass.

**Table 1 molecules-25-00541-t001:** Glycerol as the reaction solvent for extraction of chitin and chitosan.

Natural Resource	Reaction Solvent	Product	Condition
glycerol	hot glycerol	chitin	-
	Glycerol + 5–7% HCl	55 kDa chitin 57 kDa chitin	7% HCl, 120 °C, 2 h 5% HCl, 150 °C, 2 h
	Glycerol + 30% NaOH	chitosan	180 °C, 12 h, liquid–solid = 40

**Table 2 molecules-25-00541-t002:** HMF production from biomass by acid catalysts.

Substrates	Acid Catalyst	Reaction Conditions	HMF (%)	References
Chitosan	2.2% H_2_SO_4_	174 °C, 36.9 min	12.1 (wt.)	[56]
GlcNH_2_	3% H_2_SO_4_	175 °C, 5 min	1.8 (wt.)	[57]
GlcNH_2_	0.1 M MSA	160 °C, 40 min	2.3 (wt.)	[60]
Chitosan	0.1 M MSA	200 °C,15 min	15.0 (wt.)	[61]
Chitosan	0.12 mmol SnCl_4_⋅5H_2_O	MW200 °C, 30 min	10.0 (wt.)	[55]
GlcNH_2_	67% ZnCl_2_	120 °C, 90 min	21.9 (mol)	[66]
GlcNAc	FeCl_2_⋅4H_2_O	190 °C, 6 h DMSO-water solvent	33.9 (mol)	[67]
Chitosan			26.6 (mol)	
Chitosan	0.7 M NH_2_SO_3_H	200 °C, 2 min	21.5 (wt.)	[70]

**Table 3 molecules-25-00541-t003:** HMF production from biomass using ILs.

Substrates	ILs	Reaction Conditions	HMF (%)	References
Chitosan	[BMIM]HSO_4_	180 °C, 5 h 100 mol% AlCl_3_⋅6H_2_O	25.2 (mol)	[84]
Chitosan	[MIM]HSO_4_	180 °C, 5 h	29.5 (mol)	[85]
Chitin			19.3 (mol)	
Chitin	[[Hmim][HSO_4_]]	180 °C, 6 h	25.7 (mol)	[86]
Chitosan		DMSO-water solvent	34.7 (mol)	
Chitosan		180 °C, 6 h	30.8 (mol)	
Chitosan	[Hbim]Cl	180 °C, 3 h DMSO-water solvent	34.9 (mol)	[87]
Chitosan	[Hmim][HSO_4_]−0.5FeCl_2_	180 °C, 4 h	44.1 (mol)	[88]

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
