# Peer review of "Production of 5-Hydroxymethylfurfural from Chitin Biomass: A Review"

_molecules, 2020, doi:10.3390/molecules25030541_

Round 1

Reviewer 1 Report

This review covers the conversion of chitin biomass into 5-HMF. This article seems to be well-structured and organized, and discusses an interesting topic in the green chemistry which could draw attention from many researchers.

There are minor points authors need to address/correct.

Line 34. Requires ref. Line 49. Although authors provide application areas of HMF in section 4, it would be better if authors can suggest/provide several applications of HMF here. Line 52. In what respect, the chitin biomass is an ideal material for HMF production? Figure 1. Please check the resolution of some chemicals appears in this figure. Line 71. Please provide details.. unit of molecular weight, refs.. Figure 3. I understand that both GlcNAc and GlcNH2 can be materials for HMF production. Please reposition the "arrow" that can cover both GlcNAc and GlcNH2. Line 120 - 140. Please specify what kind of ILs/DESs have been used to extract chitin/chitosan.. Comparing with the section of the HMF production, the pretreatment section seems to be less comprehensive.  Not clear what pretreatment is for. Please provide more details in section 3 (line 88). Figure 4. Please redraw three chemicals. Glucose and fructose vs. HMF have a different pattern in their structure.  Please use the decimal point uniformly throughout the manuscript.  In section 4.3., some ILs have been used as either reaction media and/or catalyst. Please double-check the specific role of each IL and update manuscript and Table 383 if needed.

Reviewer 2 Report

Traditional chemical methods involve the use of diluted HCl solution and diluted sodium 105 hydroxide (NaOH) solution to dissolve the calcium carbonate and protein in the crustacean shells, respectively, to produce purified chitin. >>> This claim under describes the aggressive nature of this reaction, more details need to be provided.

Some researchers are already investigating the use of natural resources, such as glycerol, as the 142 reaction solvent for extraction of chitin. Glycerol, a by-product of biodiesel production, is readily 143 available, non-toxic, inexpensive, biodegradable, and has deproteinization ability. >> Provide conditions in a table

Overall, the manuscript would benefit from professional editing

Some discussion on the change in molecular weight of going from chitin to chitosan needs to be provided.
